# A metric for evaluating biological information in gene sets and its application to identify co-expressed gene clusters in PBMC

**Jason Bennett**[1], **Mikhail Pomaznoy**[1], **Akul Singhania**[1], **Bjoern Peters**[1,2]*

**1** Division of Vaccine Discovery, La Jolla Institute for Immunology, La Jolla, California, United States of America, **2** Department of Medicine, University of California San Diego, La Jolla, California, United States of America

* bpeters@lji.org

## Abstract

Recent technological advances have made the gathering of comprehensive gene expression datasets a commodity. This has shifted the limiting step of transcriptomic studies from the accumulation of data to their analyses and interpretation. The main problem in analyzing transcriptomics data is that the number of independent samples is typically much lower (<100) than the number of genes whose expression is quantified (typically >14,000). To address this, it would be desirable to reduce the gathered data's dimensionality without losing information. Clustering genes into discrete modules is one of the most commonly used tools to accomplish this task. While there are multiple clustering approaches, there is a lack of informative metrics available to evaluate the resultant clusters' biological quality. Here we present a metric that incorporates known ground truth gene sets to quantify gene clusters' biological quality derived from standard clustering techniques. The GECO (**G**round truth **E**valuation of **C**lustering **O**utcomes) metric demonstrates that quantitative and repeatable scoring of gene clusters is not only possible but computationally lightweight and robust. Unlike current methods, it allows direct comparison between gene clusters generated by different clustering techniques. It also reveals that current cluster analysis techniques often underestimate the number of clusters that should be formed from a dataset, which leads to fewer clusters of lower quality. As a test case, we applied GECO combined with k-means clustering to derive an optimal set of co-expressed gene modules derived from PBMC, which we show to be superior to previously generated modules generated on whole-blood. Overall, GECO provides a rational metric to test and compare different clustering approaches to analyze high-dimensional transcriptomic data.

## Author summary

Next-generation sequencing has spurred the creation of many techniques that attempt to distill large datasets down to a more manageable size without losing valuable information, more simply referred to as dimensionality reduction. We have sought to contribute to this effort by focusing not directly on dimensionality reduction but on interpreting the results

**Data Availability Statement:** The source code for the GECO R package can be found here: https://github.com/JasonPBennett/GECO The rna-seq

dataset used can be found here: https://dice-database.org/downloads.

**Funding:** Research reported in this publication was supported by the National Institute of Allergy and Infectious Diseases of the National Institutes of Health under Award Numbers U19AI118626 (BP) and U01AI150753 (BP). The content is solely the responsibility of the authors and does not necessarily represent the official views of the National Institutes of Health. The funders had no role in study design, data collection and analysis, decision to publish, or preparation of the manuscript.

**Competing interests:** The authors have declared that no competing interests exist.

of the most common technique used for dimensionality reduction of sequencing data: gene clustering. While methods to generate gene clusters have been well explored, the evaluation of cluster quality has not, i.e., answering the question "Have we made biologically significant clusters?" We have developed a metric that can be used to answer this question. Our metric incorporates prior biological knowledge about the data to determine if the clustering process was optimal by looking at how genes are grouped in gene clusters and determine if they make sense biologically. Our metric can also be used to provide a discrete range of values that indicate how to generate clusters with the highest potential biological information content. This metric can be utilized by any -omics level study to generate study-specific gene clusters while reducing the time spent validating gene clusters and improving confidence in the resultant clusters.

This is a *PLOS Computational Biology* Methods paper.

## Introduction

The bottleneck in transcriptomic analysis has shifted from the collection of data to its analysis and interpretation. Analyzing whole transcriptome data often raises the well-known "curse of dimensionality" problem of a large number of assessed variables (p) in a low number of samples (n) [1]. One approach to address this problem is to reduce a large number of measured transcript expression values to a lower number of independent variables that provide a simplified view of the transcriptome's overall state without obfuscating the biological information contained within it. This can be done by reducing the transcriptomic data gathered in a study to modules of genes [2] that show correlated expression patterns, which can then be analyzed as a group. Such study-specific analysis will ideally identify one or more modules of genes whose expression differentiates different sample groups in the study, such as those from individuals in different disease states [3].

Not all discovered gene modules will be co-expressed for reasons that are targeted by the specific study design. For example, genes on the Y-chromosome are expressed in a sex-specific fashion, and such co-expression patterns will be discerned in an unbiased module analysis. This highlights one of the complications of study-specific dimensionality reduction approaches: they can identify gene modules that are discriminating between sample sets of interest based on confounding factors–such as biological sex–and if these factors are not carefully balanced within the study sample set, they can result in false positives.

An alternative approach to deriving study-specific gene modules is to use pre-defined sets of genes. This is commonly done (at least implicitly) when using gene annotations from the Gene Ontology database [4–7] by grouping sets of genes that share the same function or act in the same pathway. This powerful approach benefits from knowledge gathered over decades of studying genes and their functions across many model organisms and provides dimensionality reduction and biological interpretation for observed gene expression patterns [8].

A downside of using gene sets based on ontology annotations is that there is still a large number of "dark matter" genes that do not have any formal functional annotations in ontologies, either because they have not been studied or because results from studying them have not been integrated into ontologies [9]. To overcome this, an alternative approach to derive sets of pre-defined genes for dimensionality reduction is to use gene expression patterns from large-

scale studies to identify sets of genes that are consistently co-expressed. The blood transcription modules (BTMs) [10] are a prominent example, which have been defined based on a meta-analysis of previous transcriptional studies of whole blood gene expression data assessed in microarrays. An updated version of the BTMs is available on bioRxiv [11]. Surprisingly, despite the BTM approach being powerful, practical, and in widespread use, we found only two other studies that adopted a similar approach to identify pre-defined gene modules and published them in [12,13]. In these, study-specific gene modules were created to investigate immune responses to Ebola and polysaccharide and conjugate vaccines for meningococcus, respectively.

The derivation of clusters in studies such as the BTM study takes advantage of large-scale transcriptomics datasets used to group genes that show similar expression patterns. One common technique for this purpose is k-means clustering. Five-decades since its introduction, k-means remains a valuable tool that reduces redundancy in data by identifying similar components and groups them together [14]. Analyzing and interpreting the resultant clusters has proven to be more challenging than generating the clusters. The criteria to determine what makes "good" or "bad" clusters currently relies on mathematical analysis like intra- and inter-cluster distances (the "elbow method" [15]) or the rate of similarity of points to their cluster centers (cluster silhouettes [16]). Both techniques offer valuable insights into the clusters' mathematical nature, but it is not clear if these metrics fully reflect what would be considered an 'optimal' set of clusters when applied to gene expression analysis. In addition to k-means, weighted gene co-expression network analysis (WGCNA) has been shown to be highly effective in partitioning genes into clusters based on network topology [2]. However, this method like k-means still lacks a quantitative method of determining if the resultant clusters are optimal.

Considering the limitations present in current dimensionality reduction techniques, specifically as it pertains to the optimal size of gene clusters, we set out to develop a metric that evaluates how much value a given partition of genes into clusters adds to answer fundamental biological questions. There are some methods that provide a measure of similarity between sets, such as the Jaccard index or a hypergeometric test. These are valuable tools for evaluating a single set of clusters, but they cannot be extended to directly compare different sets of clusters that have different numbers of clusters. Additionally, these metrics are more general and do not take advantage of the prior biological knowledge that can be leveraged when looking at gene clusters specifically. We designed a metric that would fill both gaps. To incorporate known biology, certain sets of biologically essential genes which are always co-expressed were selected and called our "ground-truth gene sets". The Y-chromosome genes will be present in all male subjects and show a distinct separation between individuals of different biological sex regardless of study design and conditions. The ribosomal protein genes are a ubiquitous transcript that must be co-expressed across a range of conditions due to the structural nature of the ribosome and its necessary role in cellular function. The immunoglobulin genes were specifically chosen to complement our test dataset which is composed of sorted immune-specific cells and thus the immunoglobulin genes will be grouped into a B-cell cell-type-specific group. Using a set of these co-expressed ground-truth gene sets, we asked if membership in the ground-truth set for a given gene can be inferred based on the composition of the presence and absence of other ground-truth genes in the same cluster. We refer to this as the GECO (**G**round truth **E**valuation of **C**lustering **O**utcomes) metric and applied it to evaluate clusters derived by k-means from a large scale transcriptomic dataset of sorted cell types found in peripheral blood mononuclear cells (PBMC) that were generated for the DICE database [17]. We show how the GECO metric can be used to determine the optimal number of clusters (k) on the DICE dataset and compare how different clusters perform on different problems. The

results also reveal the importance of a carefully chosen ground-truth set(s) to match the context of the biological question of interest.

## Results

### Establishment of the GECO metric

We developed the GECO (**G**round truth **E**valuation of **C**lustering **O**utcomes) metric to measure how well a set of gene clusters separates genes with a known shared 'ground truth' property from others. We wanted to ensure that the metric can be compared across different sets of gene clusters independent of the number of clusters, which is not the case for established 'information content' based metrics that we initially considered. Our approach is summarized in (**Fig 1**). The starting point is a given ground truth set of genes and a given subdivision of genes into clusters that are being evaluated. For each gene in each cluster, we assigned a score that indicates how likely that gene is to be one of the ground truth genes if all we knew about it is its cluster membership, the composition of other genes in its cluster, and the number of ground truth genes in the remainder of the data (see Methods **S1 Fig**). We then evaluated these gene scores' ability to separate ground-truth genes from others in a receiver operator curve (ROC) analysis, where the resulting area under the curve (AUC) value is the final GECO metric. As with any AUC value, a GECO metric of ~0.5 is equivalent to random assignments, while a value of 1.0 would be the result of perfect separation of the ground truth genes into clusters separate from all other genes.

### Using GECO to establish an optimal k-value for co-expressed gene clusters in the DICE database

We wanted to determine the number of biologically relevant gene clusters that can be derived when performing co-expression analysis of RNA-seq data from the DICE database [17]. The DICE dataset consists of 15 immune cell types obtained by sorting frozen PBMC samples from healthy individuals. We used k-means clustering to identify sets of co-expressed genes within this dataset. Initially, the popular "elbow method" was used to identify the optimal number of clusters. This method relied on plotting the within-sum-of-squares (WSS) values for a range of k's and choosing a k-value where the WSS began to level off, resulting in an optimal k-value of 16 (**S2 Fig**).

Next, we wanted to determine which k-value is considered optimal based on the GECO metric for three ground truth sets of genes that are expected to be co-expressed based on their shared biology, namely i) Ribosomal protein genes, ii) Immunoglobulin genes, and iii) Sex-specific (Y chromosome) genes (**S1 Table**). The co-expression of these ground truth gene sets in the DICE dataset was confirmed based on their expression pattern in a heatmap (**S3 Fig**).

Using k-means clustering of the DICE genes, k = 16 clusters were generated (as recommended based on the elbow method), and ten iterations were performed to account for the stochasticity in initial cluster conditions. The resulting clusters were scored using the ribosomal protein gene ground-truth set with the mean GECO metric across all ten iterations being 0.845 (**Fig 2A**). In comparison, a random control set of genes picked from the DICE dataset was also tested. This random control set contained the same number of genes as the ribosomal protein gene ground truth set and excluded any genes found in the true ribosomal ground truth set by sampling without replacement from all remaining genes in the dataset. Due to the random assignment, these genes are not expected to be significantly co-expressed and not expected to be readily separable from other genes in k-means clustering. This was indeed the case with an average GECO metric of 0.483. Overall, this supports that k-means

## Partition genes into clusters

## Assign each gene in the cluster a score

$g$ = the current gene

 $g$=1 if $g$ is ground truth gene

 $g$=0 if $g$ is not a ground truth gene

$$b = \frac{\text{total number of ground truth genes} - g}{\text{total number of genes in dataset} - 1}$$

Gene Score:

$$\frac{\frac{\text{Number of ground truth genes in cluster } - g}{\text{Number of total genes in cluster} - 1}}{b}$$

Regular Gene 1

**Ground Truth Gene 1**

Regular Gene 2

 Regular Gene 3

**Ground Truth Gene 2**

Regular Gene 4

 Regular Gene 5

**Ground Truth Gene 3**

Regular Gene 6

## Score all genes in all clusters

| Cluster Label | Gene Name | Gene Score |
|---|---|---|
| "green" | Regular Gene 1 | x |
| "green" | **Ground Truth Gene 1** | y |
| "green" | Regular Gene 2 | x |
| … | … | … |

## Use gene scores to calculate the GECO metric for the clusters

**ROC Plot**

True positive rate

GECO metric: …

False positive rate

**Fig 1. The GECO metric scoring process; from cluster assignment to GECO score.** Transcriptomic data was clustered using k-means clustering. Each cluster contained both ground truth genes (seen in blue) and non-ground truth genes (seen in grey). The scoring function applied to each gene in each cluster. The gene score reflects the likelihood of that gene being a ground truth gene based on the makeup of the cluster to which the gene belongs and the distribution of ground truth genes throughout the dataset. A table containing all the genes in the dataset, scored by cluster, and their associated gene scores. The gene scores are used to generate a ROC plot and the corresponding AUC value is the GECO metric which indicates the overall quality of the clusters.

clustering with k = 16 provides groups of genes that separate ribosomal proteins from others and that the GECO metric separates meaningful from random gene cluster assignments.

 We repeated this process for a comprehensive range of k-values ranging from a k-value of 1, indicative of all genes being in one cluster, to k = 14,175 indicative of each cluster consisting of a single gene ("singlet" clusters), using geometric increases in k. Contrary to what was

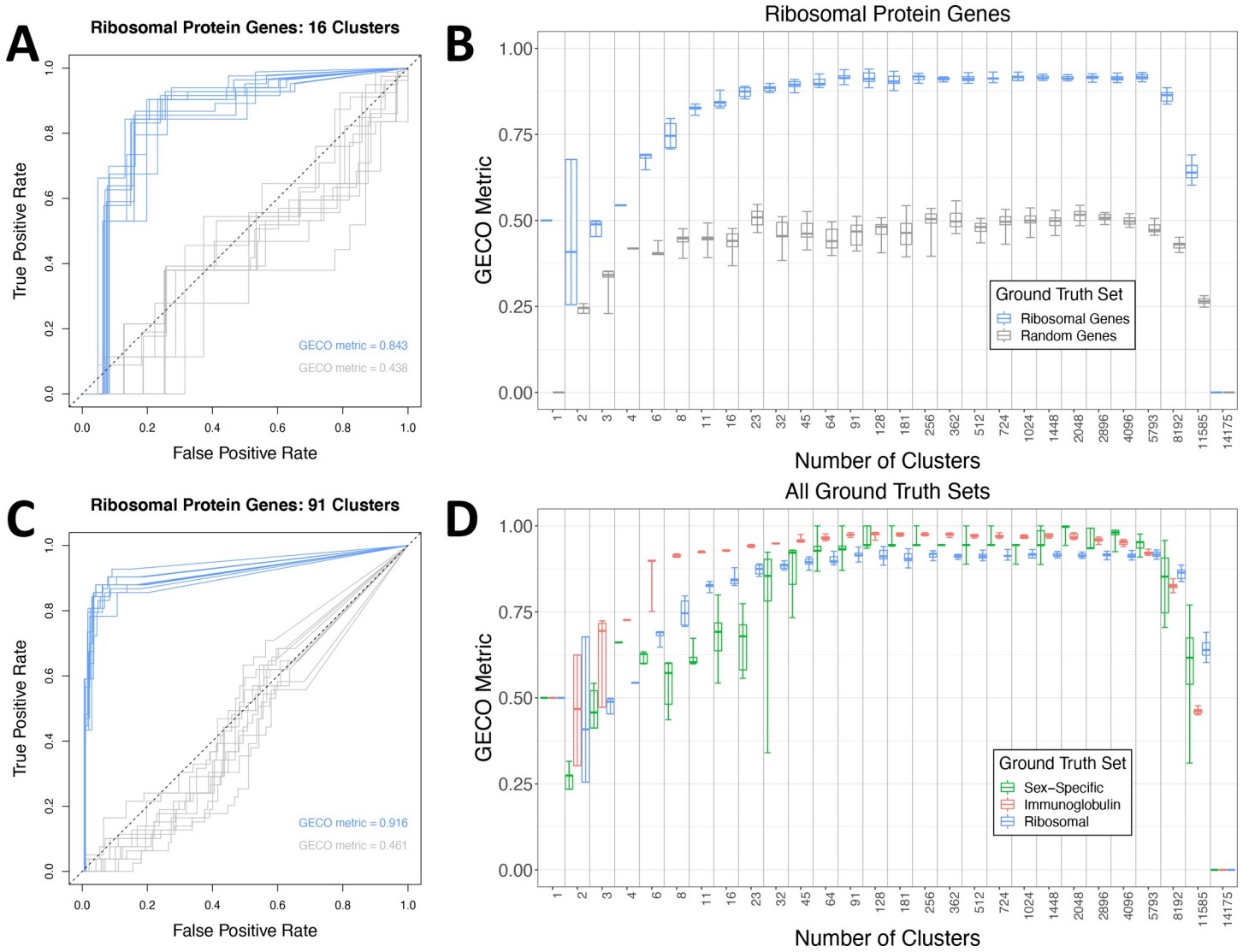

**Fig 2. Using the GECO metric to determine an optimal number of clusters.** *A*. A ROC plot using the scores from the GECO metric over ten iterations of k-means clustering with 16 clusters. In blue are the GECO scores for the ribosomal protein gene ground truth set, in grey are the scores for an equivalent number of randomly sampled genes. The GECO metric value is also noted for each group of genes. *B.* K-means clustering was performed ten times for each value of k ranging logarithmically from 1–14,175; each cluster iteration was scored using the GECO metric. The values for the ribosomal protein gene ground truth set are plotted in blue, while the grey scores are the values for an equivalent number of randomly sampled genes. The boxes cover the 0.25 to 0.75 confidence interval. The whiskers range from minimum to maximum values. The bar within the boxes indicate the mean value over the ten iterations. *C.* A ROC plot generated similar to A, but with 91 clusters. Blue represents the ribosomal protein gene ground truth set and grey the randomly sampled genes. The average GECO metric values are again provided. *D.* All three ground truth sets are plotted after their GECO metric scores are calculated. The boxes, whiskers, and inner bars are used in the same manner as detailed in section 2B.

indicated by the elbow method, increasing k-values above 16 resulted in a marked improvement in cluster quality scores up to a k-value of 91, after which the scores plateaued and dropped off for very high values of k > ~8,000 (**Fig 2B**). At k = 91, the GECO metric indicated high cluster quality, with 70% of the co-expressed ribosomal protein genes found grouped in two clusters and showed a distinct separation from the control ground truth set (**Fig 2C**). This process was repeated with each ground truth gene set (**Fig 2D, S2 Table**), and the k-value of 91 showed a significant improvement compared to the k-value of 16 favored by the elbow method for all ground truth sets (**Figs 2D and S4**). While the ribosomal protein gene ground

truth set indicated that a k = 91 was optimal, it was not confirmed as the optimal k-value until we included the other ground truth gene sets.

Next, we used the Weighted Gene Co-expression Network Analysis (WGCNA) package in R [2] to identify co-expressed genes as an alternative to k-means clustering and to demonstrate the utility of the GECO metric independent of the clustering method used. Depending on the 'deep split' variable chosen, WGCNA identified optimal values of k = 26, 36, 46, and 49. We tested the clusters generated using WGCNA on the GECO metric, and they provided comparable scores to clusters generated by k-means at their respective k-values (**S3 Table**). This supports that WGCNA provides a meaningful grouping of genes in clustering, but it suggests that the number of clusters identified could be further increased–which is not straightforward to parameterize in a WGCNA run, which is why we chose k-means here as a baseline clustering approach. Using the GECO metric, we found that the optimal number of biological clusters in the DICE dataset is much higher than that identified by popular clustering techniques such as k-means and WGCNA.

## Choice of ground truth data and the context of its application impacts GECO scores

Next, we wanted to evaluate how the dataset used to generate the clusters and the source of the ground truth gene sets chosen for scoring those clusters affected the GECO metric's output.

We evaluated the DICE clusters once again and asked how well the Burel 74-gene signature [18] distinguishing CD4 memory T cell gene expression in latently Tuberculosis (TB) infected from non-infected individuals could be separated by the DICE gene clusters. As expected, the cluster quality score for the Burel signature was lower than the scores for the other ground truth sets, as the DICE dataset is composed of healthy individuals (**Fig 3A**). Significant co-expression of the latent TB signature CD4 signature genes was not present in the DICE clusters, and as a result, this gene set did not drive cluster formation resulting in the omission of a clear TB signature gene cluster and a lower score from the GECO metric.

Next, we used the "Burel" gene expression dataset gathered from CD4 T cells of individuals with different statuses of TB infection (see Methods) and performed k-means clustering to generate a new set of "Burel clusters." We asked how well these clusters were able to separate different ground truth gene sets from others. We considered the original ribosomal protein and sex-specific ground truth gene sets as they should remain strongly co-expressed within the Burel CD4 T cell dataset, but not the immunoglobulin genes, as they are not expressed in CD4 T cells. In addition, we added the 74-gene TB signature as an expected co-expressed gene set. As expected, the Burel clusters had higher GECO scores when tested for their ability to identify the Burel TB signature as a ground truth dataset compared to the DICE data (**Fig 3B**). The ribosomal and sex-specific ground-truth gene sets reported high cluster quality scores within the Burel clusters compared to the DICE clusters. Further, the Burel clusters' optimal k-value was lower than that in the DICE clusters, with a peak identified at a k of 64 (**Fig 3B**). Overall, these data suggest there is no universal optimal k-value and that the GECO metric provides the most meaningful interpretation when the query data set and ground truth set(s) are derived from a similar biological context.

## The final set of 'k91 DICE clusters' captures a breadth of immune responses

As described above, a k-value of 91 was found to be an optimal number of clusters for the DICE dataset when averaging over ten iterations. We performed pairwise comparisons between all genes in each cluster and each iteration to determine which iteration was most

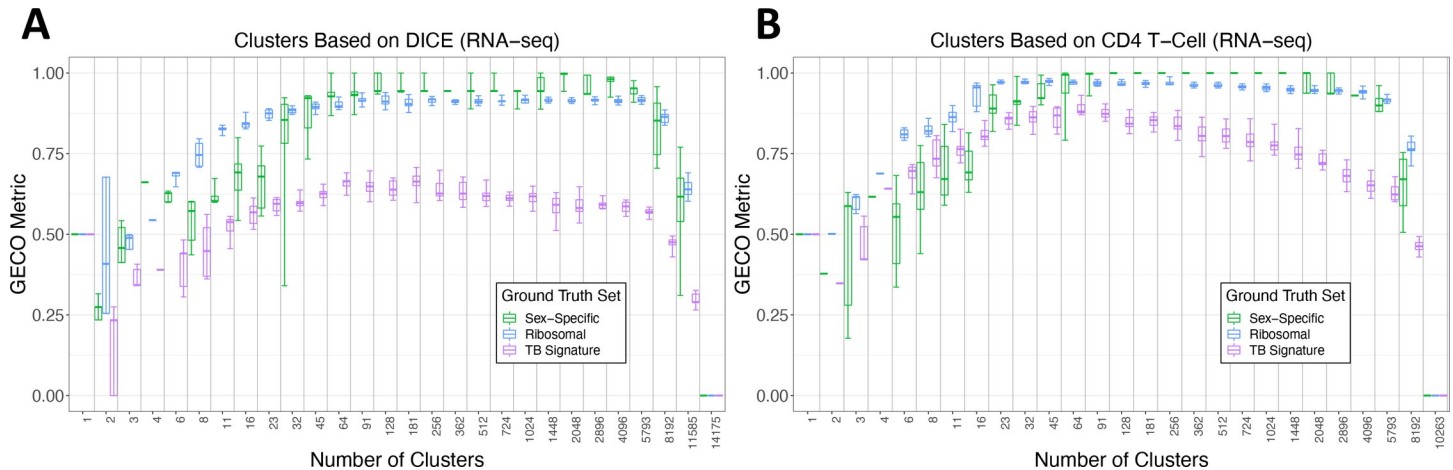

**Fig 3. GECO performance is driven by the choice of the ground truth dataset.** *A*. K-means clustering performed on the 14,175-gene DICE dataset ten times per number of clusters (the x-axis value). Each iteration was scored by GECO, the scores were used to generate ROC plots, and those plots produced AUC values which are represented in the boxes in the boxplot. The boxes cover the 0.25 to 0.75 confidence interval. The whiskers range from minimum to maximum values. The bar within the boxes indicate the mean value over the ten iterations. The purple datapoints are scores from the "Burel" 74-gene CD4 T-cell TB signature. *B*. K-means clustering and GECO scoring of the 10,263-gene CD4 T-cell dataset carried out in the same manner as part A. The scores for the Burel 74-gene CD4 T-cell TB signature are significantly higher in the CD4 T-cell dataset compared to the DICE dataset due to the sensitivity of the metric to the choice of ground truth set in relation to the dataset.

representative of the ten total iterations. This was determined by counting the number of times each gene pair within each cluster was also found clustered together within the other nine iterations. The iteration with the highest count of co-occurring gene pairs was selected as the final 'k91 DICE cluster' set listed in **S4 Table**. The concentration of ground truth genes within the k91 DICE clusters can be seen in **S5 Table**. We also compared the k91 DICE clusters to the default set of WGCNA clusters (the ones generated without varying the 'deepSplit' parameter) and found significant overlap between the clusters generated by the two methods (**S5 Fig**). Using functional enrichment analysis, we identified the biological and immunological functions associated with each of the clusters within the k91 DICE clusters. We identified clusters associated with monocytes (classical and non-classical) enriched for the phagocytosis biological process (Clusters 7, 9, and 49). There were clusters associated with CD4 and CD8 stimulated T cells that showed enrichment in ribosome biogenesis and mitochondrial expression (Clusters 57, 80, and 86). There was another cluster involved in leukocyte production and adhesion and associated with stimulated CD4 T cells (Cluster 8) and a cluster associated with $T_h2$ cells that showed enrichment in interleukin-5 production and regulation. We also identified a cluster associated with antigen presentation. However, this cluster was not enriched for any one particular cell type (Cluster 2). A single cluster (Cluster 12) contained ten genes in total and included all nine genes from our sex-specific ground truth set as well as a gene that was previously unknown to us called ANOS1 (previously ADMLX) which plays a crucial role in Kallmann Syndrome, an X-linked disorder [19]. In sum, the k91 DICE clusters represent and capture a breadth of immune responses and offer a unique tool to evaluate the immune responses perturbed in cell-type-specific disease datasets.

## Evaluation of k91 DICE clusters using molecular signatures database

We also scored the k91 DICE clusters using the molecular signatures database (MSigDB) hallmark gene sets [20] (**S6 Table**). The hallmark gene sets contained 50 different gene signatures used for gene set enrichment analysis. The GECO metric has identified multiple hallmark gene sets that are well captured by the k91 DICE clusters. The top two hallmark gene sets are

the 'HALLMARK_MYC_TARGETS_V2' and 'HALLMARK_MYC_TARGETS_V1' gene sets with GECO metric values of 0.86 and 0.82 respectively. These contain genes that are regulated by the proto-oncogene Myc. For both gene sets, two k91 DICE clusters (Clusters 17 and 80) contained the majority of the hallmark genes. These two clusters were associated with CD4 and CD8 stimulated T cells and were shown through GO enrichments to contain genes associated with ribosome biogenesis. Additionally, k91 DICE cluster 17 was shown through a KEGG pathway enrichment to be enriched in genes associated with 'Human T cell leukemia virus I infection'. We were also able to identify another hallmark gene set, called the 'HALLMARK_INFLAMMATORY_RESPONSE', which was successfully captured by the k91 DICE clusters with a GECO metric value of 0.74. This gene set was found split primarily between three k91 DICE clusters (Clusters 28, 9, and 23) and all three clusters were strongly associated with monocytes which follows the known biology linking monocytes to inflammation. Investigating the lowest scoring hallmark gene sets, we saw the 'HALLMARK_SPERMATOGENESIS' and 'HALLMARK_ANGIOGENESIS' sets. We would not expect these gene sets to be significant captured by our k91 DICE clusters, as these processes are not associated with immune cells in the blood. Investigating the MSigDB hallmark gene sets both increased our confidence in the biological significance of the k91 DICE clusters as well as reinforced the accuracy of the GECO metric. In addition, the MSigDB hallmark gene sets are not mutually exclusive–many genes are shared between different hallmark gene sets but the metric has no issue evaluating these non-mutually exclusive gene sets.

## Evaluation of k91 DICE clusters against other established cluster sets

We next evaluated the performance of the k91 DICE clusters against the previously established blood transcription modules (BTMs) [10,11]. These modules are commonly used in the immunology field to evaluate transcriptomic disease datasets. The performance of how well the Burel T cell TB signature is co-expressed within the identified clusters was evaluated in the k91 DICE clusters and the BTMs (**Fig 4A**). The k91 DICE clusters showed superior performance in capturing the Burel CD4 T cell TB signature's co-expression compared to the BTMs (**Fig 4A**).

As the DICE clusters were identified using the DICE dataset, which is composed of discrete cell types including memory CD4 T cells, and as the BTMs were identified using whole blood datasets, we evaluated a 393-transcript whole blood TB signature [21]. The whole blood TB signature was identified more readily in the BTMs than in the DICE clusters (**Fig 4B**). Of note, the newer 2018 BTMs offered only a slight improvement in performance detecting the whole blood TB signature compared to the original 2008 BTMs (**Fig 4B**). These results indicate that the DICE clusters perform better when evaluating cell-type-specific gene expression, whereas the BTMs show improved performance for whole blood gene expression. Overall, this demonstrates the usefulness of having the GECO metric evaluate how different gene set clusterings perform for specific biological questions.

## Discussion

Partitioning genes into clusters is a common, widely used dimensionality reduction technique. The two most common methods of gene clustering are k-means and WGCNA. While k-means is lightweight, robust, and broadly applicable across multiple disciplines, it lacks the sensitivity required for biological applications when used by itself. WGCNA was purpose-built for biological clustering and produced much better gene clusters on its own compared to simple clustering with k-means (**S3 Table**: k-means "elbow" vs. any WGCNA clusters). While the clusters generated by WGCNA are known to be of higher quality due to post hoc analysis

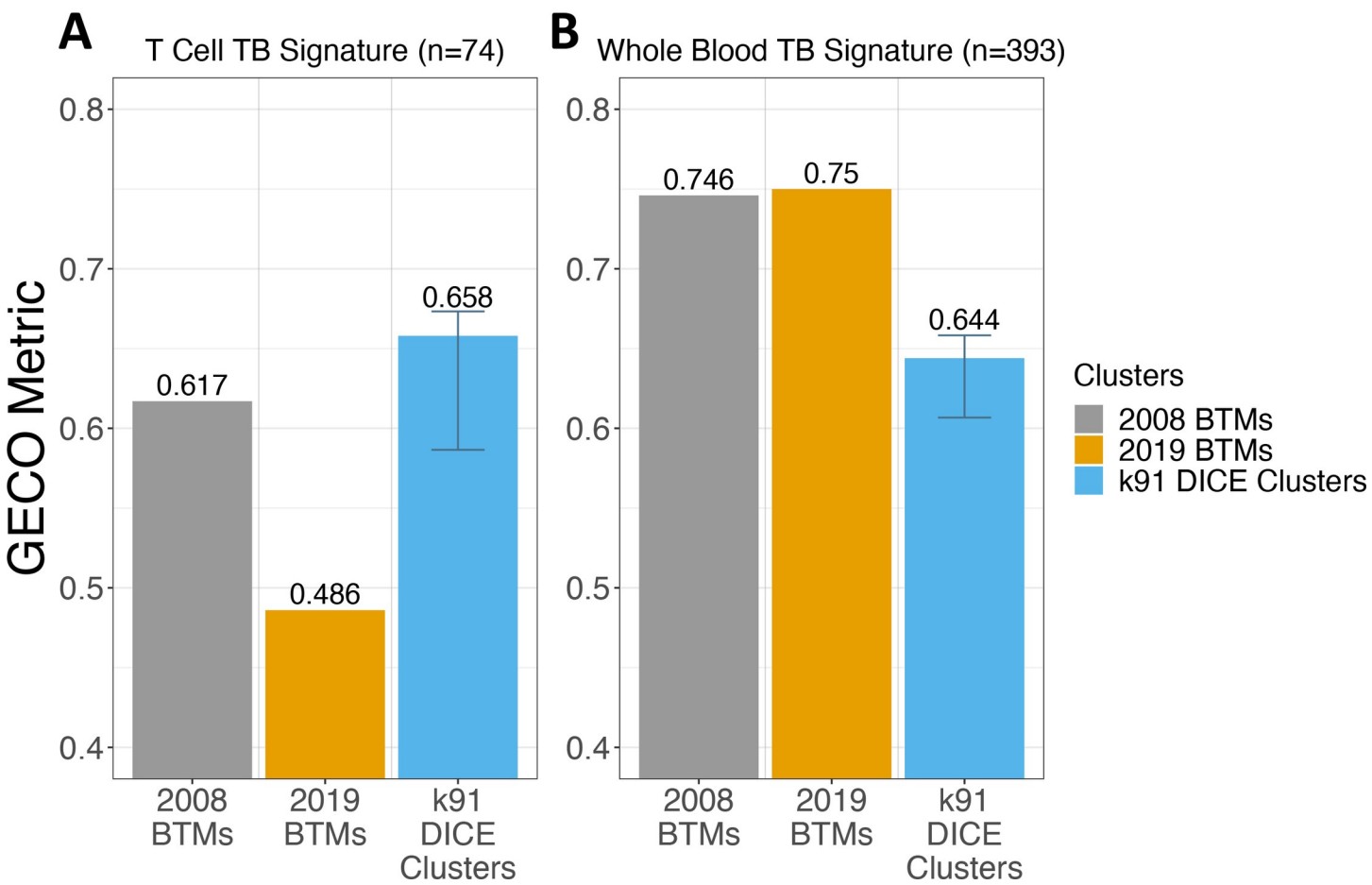

**Fig 4. Validating the metric using known gene modules and the k91 DICE clusters. *A.*** The GECO metric values evaluating the cluster quality for three sets of clusters; the 2008 BTMs, the 2019 BTMs, and the k91 DICE clusters. The ground truth set used to score the different cluster groups was the "Burel" 74-gene CD4 T-cell TB signature. The error bar on the k91 DICE clusters indicate the range of potential scores from the other 9 iterations of the k91 DICE clusters. ***B.*** The GECO metric values evaluating the cluster quality of the same three cluster sets as before. The ground truth set used to score the clusters was the "Berry" 393-gene whole blood TB signature.

using gene set enrichment, the actual level of improvement is difficult to quantify. To this point, no methods existed that allowed for precise measurements of the biological quality of gene clusters. The GECO metric fills this gap and allows users to compare k-means clusters to WGCNA clusters and others.

Using the DICE dataset, we created a set of gene clusters using WGCNA and compared them to the clusters generated by k-means using the "elbow method" to select our k-value. The WGCNA clusters showed increased enrichment of all three ground truth gene sets compared to the k-means clusters. Using the number of clusters generated by WGCNA as our new k-value, k-means clustering was repeated and produced comparable clusters to WGCNA again as determined by the enrichment of ground truth genes (**S3 Table**). This indicated to us that when used alone, WGCNA produced more biologically informative clusters than k-means clustering. However, k-means clustering could be used to generate high-quality clusters rivaling WGCNA if prior biological knowledge was incorporated into the clustering process (**S3 Table**: GECO vs. any WGCNA clusters). This is what the GECO metric calculates; given known co-expressed gene sets as ground truth, how well are these gene sets partitioned in the current clusters.

There are some potential drawbacks related to the selection of ground truth gene sets. The ground truth genes lie at the heart of the metric, meaning that selecting poorly understood

gene sets or gene sets that do not fit the data will result in inaccurate measurements from the GECO metric. To avoid this, users must ensure that the biological context of the data matches that of the ground truth sets. Additionally, it is possible to use the metric with only a single ground truth set. This would produce highly biased results that overemphasize the effect a single ground truth set has on the clustering. Multiple ground truth sets need to be included, and while some ground truth sets may be given more weight when determining the optimal clustering configuration (such as a disease specific signature used on a disease specific dataset), a single ground truth set should not be considered sufficient to make a final determination. We suggest the following methods for creating ground truth sets; either performing a rigorous literature search to select signatures from peer reviewed sources or using dataset-specific disease signatures identified by differential expression analysis (where applicable).

Using GECO with an iterative k-means approach allowed us to identify an "optimal" k-value that would produce the highest quality clusters, as determined by an increased enrichment of ground truth genes. Using this approach on the DICE dataset, we identified a k-value of 91, much larger than any k-value obtained by either the "elbow method" or WGCNA (**Fig 2D**). The k-value of 91 represents one value from a range of potential values that could be considered "optimal." GECO does not produce a single best value for k. Rather, it provides a range within which higher-quality clusters are guaranteed. The 91 DICE clusters identified using GECO were confirmed to be rich in biological informational content through gene set enrichment using all GO ontologies and the identification of cell-type-specific clusters that allowed us to supply annotations for 63 of the 91 total clusters (**S7 Table**).

We compared the k91 DICE clusters to existing, high-quality gene clusters referred to as the blood transcription modules (BTMs). These modules have been shown to identify disease states in studies using whole blood transcriptional analysis [22–24]. In a head-to-head comparison of the k91 DICE clusters against the BTMs, the k91 DICE clusters were determined to be much higher quality than the BTMs when attempting to capture the "Burel" 74-gene CD4 T-cell tuberculosis signature (**Fig 4A**). However, when using the "Berry" 394-gene whole blood tuberculosis signature, both versions of the BTMs were assessed as higher quality than the k91 DICE clusters (**Fig 4B**). Both of these findings reinforced our hypothesis that the BTMs would excel at identifying biological information contained in whole blood transcriptomic analyses, whereas the k91 DICE clusters would be especially suited for use with PBMC data.

This finding underscored the value in creating study-specific clusters, as careful consideration regarding the nature of the data plays an integral role in developing high-quality clusters. Non-study-specific gene modules that are based largely on ontological classifications will overlook uniquely upregulated genes observed only under specific study conditions. Despite this, few other studies could be found that attempted to create their own gene clusters like the BTMs or the k91 DICE clusters. A significant bottleneck of this approach was the inability to assess the quality of the clusters created in a time-efficient manner. With GECO, clusters can be generated repeatedly and quickly assessed, allowing for specialized, high-quality clusters to be produced, leading to improved dimensionality reduction without loss of fine-grain biological information. Further, GECO allows direct comparison between clustering results which had previously been a labor-intensive and time-consuming task.

## Methods

### k-means clusters derived from DICE transcriptomic dataset

The DICE database (Database of Immune Cell Expression, Expression quantitative trait loci (eQTLs) and Epigenomics [17]) contains RNA-seq data from PBMC samples collected from 91 healthy individuals separated into 15 cell types: classical monocytes, non-classical

monocytes, NK cells, naïve B cells, naïve CD4 T cells, naïve CD8 T cells, activated naïve CD4 T cells, activated naïve CD8 T cells, naïve $T_{reg}$ cells, memory $T_{reg}$ cells, $T_h1$ cells, $T_h1/17$ cells, $T_h17$ cells, $T_h2$ cells, and $T_{fh}$ cells. We retrieved data for 86 donors where all 15 cell types were analyzed. Raw RNA-seq data was TPM (Transcripts Per Million) normalized. Next, lowly expressed genes that failed to meet an average expression threshold of 1 TPM in at least one cell type were removed. Z-score normalization was then performed on the TPM normalized and filtered counts in a gene-wise fashion. Additional filtering was performed to remove non-protein-coding genes, resulting in a total of 14,175 genes across 1,290 samples (15 cell types from 86 donors).

K-means clustering was applied to the filtered dataset using the k-means function in R (v. 3.6.2) with the following parameters: algorithm = "Hartigan-Wong", nstarts = 1, iter.max = 40, and a range of k values sweeping from 1 to 14,175 increasing in logarithmic steps (14,175 being the total genes in the dataset). For the k-means algorithm, the distance between genes was calculated using the TPM normalized z-scores and finding the geometric distance between the genes using the Euclidean distance metric.

## Ground truth data

We compiled three "ground truth" sets of genes. Genes within each set are known to be co-expressed based on shared biological mechanisms. Specifically, the sets selected (listed in **S1 Table**) were:

1. Ribosomal protein genes: Genes were selected by using the HGNC "Ribosomal Protein" gene group [25]. Mitochondrial ribosomal genes and non-protein-coding genes were removed. The final ribosomal ground truth set contained 83 genes.

2. Immunoglobulin genes: Genes were selected by using the HGNC "Immunoglobulin (IGs)" gene group [25]. Non-protein coding genes were removed. The final immunoglobulin ground truth set contained 155 genes.

3. Sex-specific (or Y chromosome) genes: Genes were selected based on being encoded on the Y chromosome. Non-protein coding genes were removed. The final Y chromosome ground truth set contained nine genes.

## GECO metric

The GECO metric will assign each gene a score based on the size of that genes' cluster (one gene or multiple genes in the cluster) and the identity of the gene (either the gene is in a ground truth set or it is not) (**Fig 1**).

For clusters that had more than one gene member, each gene could either be a ground truth gene or a non-ground truth gene. The scores for both cases were calculated as follows:

1. Ground Truth Gene: The score was calculated by taking the total number of ground truth genes found within the cluster– 1 divided by the total number of genes found within the cluster– 1. This ratio is then divided by the background frequency of ground truth genes, defined by taking the total number of ground truth genes in the dataset– 1 divided by the total number of genes in the dataset– 1.

$$\frac{\frac{\#\ of\ ground\ truth\ genes\ in\ the\ cluster-1}{size\ of\ the\ cluster-1}}{\frac{\#\ of\ ground\ truth\ genes\ in\ total-1}{total\ number\ of\ genes\ in\ dataset-1}}$$

2. <u>Non-ground Truth Gene</u>: The score was calculated by taking the total number of ground truth genes found within the cluster divided by the total number of genes found within the cluster– 1. This ratio is then divided by the background frequency of ground truth genes, defined by taking the total number of ground truth genes in the dataset divided by the total number of genes in the dataset– 1.

$$\frac{\frac{\#\ of\ ground\ truth\ genes\ in\ the\ cluster}{size\ of\ the\ cluster-1}}{\frac{\#\ of\ ground\ truth\ genes\ in\ total}{total\ number\ of\ genes\ in\ dataset-1}}$$

For clusters with only a single gene member (singlet), each gene could either be a ground truth gene or a non-ground truth gene. The scores for both cases were calculated as follows:

3. <u>Singlet, Ground Truth Gene</u>: The score was calculated by taking the total number of ground truth genes from the entire dataset– 1 divided by the total number of genes in the dataset– 1 (the background frequency or probability that any gene selected at random could be a ground truth gene).

$$\frac{\#\ of\ ground\ truth\ genes\ in\ total - 1}{total\ number\ of\ genes\ in\ dataset - 1}$$

4. <u>Singlet, Non-ground Truth Gene</u>: The score was calculated by taking the total number of ground truth genes from the entire dataset divided by the total number of genes in the dataset– 1.

$$\frac{\#\ of\ ground\ truth\ genes\ in\ total}{total\ number\ of\ genes\ in\ dataset - 1}$$

For the gene scores derived from singlet clusters, nothing about the gene can be learned from other cluster members. So, the scores were calculated based on the estimated likelihood of the gene being a ground-truth gene based on the frequency of ground-truth genes in the entire dataset. This is the reason two different methods of scoring needed to be derived.

The genes' scores were used to generate receiver operator curves using ROCR (v. 1.0–11) and to calculate an AUC value for the retrieval of the ground truth genes using these scores. The AUC value for a given set of gene clusters and a given set of ground truth genes is the GECO metric.

The GECO metric was established to provide a measure of how well a set of gene clusters separates genes from a given ground-truth dataset from other genes. This was done by inspecting each cluster and identifying the number of ground truth genes found within, then subsequently assigning each gene within that cluster a score based on the distribution of ground truth genes found. The score for each gene represented the likelihood that the gene was a member of the ground truth set if nothing else was known about the gene apart from it being a member of its cluster, what the other members of the cluster are, and what the frequency of ground truth genes was in the remaining dataset (referred to as the 'background frequency' of ground truth genes). Such scores ranged from 0.0 in the case where no ground truth genes existed within the cluster up to a potential maximum of 1.0, where all other genes in the cluster were ground truth genes. Since the gene scores are simply predictions, ROC curves were used to calculate the effect different cutoffs (or different partitions of ground truth genes) had on the ability to predict how successfully the clusters had captured known co-expressed gene sets.

### Generation and annotation of the 'k91 DICE clusters'

As described in the results, a k-value of 91 proved optimal for the DICE dataset when results were averaged over ten independent iterations. To select a robust representative cluster, the ten iterations were compared against each other by taking each pair of genes from each cluster in a single iteration and checking how often that pair was clustered together in any of the other nine iterations. The overall score for an iteration was the sum of all gene pair scores; the iteration with the highest score was chosen as the final set of clusters. This set of clusters was called the 'k91 DICE clusters' (**S4 Table**).

The k91 clusters were functionally annotated using data from GO, KEGG, and DICE to assign putative shared roles to the genes in each cluster [4,6,17]. Gene ontology (GO) enrichment was performed using the enrichGO function from clusterProfiler package (v. 3.14.3) in R. The parameters used for the enrichGO function were as follows: OrgDb = org.Hs.eg.db (v. 3.11.4), keyType = "ENSEMBL", ont = "BP,MF,CC", pAdjustMethod = "BH", pvalueCutoff = 0.01, and qvalueCutoff = 0.05. The background gene set for the enrichment contained all the genes in our dataset. The simplify function in the clusterProfiler profiler package was then used to reduce semantic similarity between the significantly returned GO terms, and the parameters used were as follows: cutoff = 0.7, by = "p.adjust", select_fun = min, measure = "Wang". The top 10 most significant, non-redundant GO terms were reported as annotations for each of the k91 DICE clusters. KEGG pathway enrichment was performed using the enrichKEGG function from clusterProfiler. The genes were converted to the NCBI naming convention and the parameters used in enrichKEGG were: organism = "hsa", keyType = "ncbi-geneid", pAdjustMethod = "BH", pvalueCutoff = 0.01, and qvalueCutoff = 0.05. As with the GO enrichment, the background gene set for the enrichment contained all the genes in our dataset. Additionally, cell type enrichment was also performed using the DICE Tools application CellTypeScore [26]. Each cluster was fed into the tool in the form of a list of HGNC gene names. CellTypeScore then summed the TPM expression of each gene within each cell type in the cluster. Cell type specificity was determined by identifying clusters with expression at least two-fold greater in a particular cell type compared to the average expression in all the other cell types.

### Generation of WGCNA gene modules

Weighted Gene Co-expression Network Analysis (WGCNA) was used on the TPM and Z-score normalized reads to generate gene modules to compare against the k-means modules. The parameters used to generate the blockwise modules for WGCNA were: corType = "pearson", maxBlockSize = 15000, networkType = "unsigned", power = 10, minModuleSize = 9, and mergeCutHeight = 0.25. We used the 'pickSoftThreshold' function with a range of powers from 1–20. Based on graphs of the scale-freeness and mean connectivity, a power of 10 was selected (**S6 Fig**). Four iterations of WGCNA were run with the above parameters while updating the deepSplit = 1, 2, 3, and 4 and one iteration was performed using the default value for the deepSplit parameter. All four non-default iterations were retained to be compared to naïve k-means clusters as well as the k-91 DICE clusters.

### Generation of gene clusters from the Burel dataset

A second publicly available RNA-seq dataset [18] was used to validate the GECO metric. The "Burel" dataset contains MTB300-stimulated [27] sorted memory CD4 T cells from 59 donors (29 control subjects and 30 individuals with latent tuberculosis infection). As with the previous dataset, raw RNA-seq data was TPM normalized, and gene filtering was performed to remove genes that failed to reach a TPM threshold of 1 in any sample. Additionally, non-protein-

coding genes were removed, leaving a total of 10,263 genes. Gene names were converted from UCSC gene names to ENSEMBL gene names. Clusters were generated using k-means clustering of the Burel dataset, using the same parameters as described above. Clusters were generated for k values ranging from 1 to 10,263, increasing in logarithmic steps (10,263 being the total genes in this dataset).

### Compilation of disease-specific ground truth gene sets

Two additional ground-truth gene sets were compiled to evaluate GECO in the context of a disease setting: the Burel 74-gene memory CD4 T cell latent tuberculosis signature [18] identified in the second RNA-seq validation dataset above, and the Berry 393-gene whole-blood active tuberculosis signature [21]. They were filtered to remove non-protein-coding genes, and any incompatible gene names were either converted into ENSEMBL gene names or dropped if the conversion was not possible (S8 Table).

In addition, two sets of blood transcription modules (BTMs) that were derived from whole blood and developed by Chaussabel et al. [10,11] were used. The BTMs generated in 2008 consist of 7,241 genes separated into 346 modules [10], while the more recently established BTMs generated in 2018 consist of 14,504 genes separated into 382 modules [11]. Non-protein coding genes were removed from both sets. Some gene names from the 2008 BTMs were reported with non-alphanumeric characters included; these were excluded from the modules, resulting in 7,179 genes remaining in the 2008 BTMs.

### Supporting information

**S1 Fig. The GECO metric scoring process; from cluster assignment to calculated scores. *i*.**) The GECO metric requires a list of genes, a label indicating to which cluster each gene has been assigned, and a Boolean value indicating if the gene is a ground truth gene. ***ii*.**) Each gene is then assigned a gene score based on the likelihood of that gene being a ground truth gene based on the makeup of the cluster to which the gene belongs. ***iii*.**) Two sample genes scored, one a ground truth gene and one a regular gene, to illustrate the scoring process. ***iv*.**) The gene scores are stored to later be used in the final calculation of the GECO metric.
(TIF)

**S2 Fig. The elbow method evaluation of the DICE dataset.** The results of the "elbow method" to determine the optimal number of clusters. A k-value of 16 was chosen based on inspection of the graph, however, no defined inflection point could be definitively identified.
(TIF)

**S3 Fig. Evaluation co-expression of ground truth genes.** The co-expression of our three ground truth gene sets as visualized using a heatmap. The values used are RNA-seq expression levels after TPM and Z-score normalization. A range of values, from -3 to 3, were used to visualize the heatmap with any values outside of that range reassigned to the nearest threshold. The ground truth sets are indicated along the y-axis. Red indicates positive scores while blue indicates negative scores.
(TIF)

**S4 Fig. k = 16 vs. k = 91 GECO quality results.** ROC plots showing the increase in cluster quality for the given ground truth gene set when increasing from k = 16 to k = 91. The GECO metric for each set is shown in their respective color and represent the mean cluster quality score. ***A***. Sex-specific ground truth gene set: Each green line represents the GECO score for a single iteration at the given k-value and the grey lines are iterations with random genes selected and scored as pseudo sex-specific genes. ***B***. Immunoglobulin ground truth gene set:

Each red line represents the GECO score for a single iteration at the given k-value and the grey lines are iterations with random genes selected and scored as pseudo immunoglobulin genes. **C**. Ribosomal Protein ground truth gene set: Each blue line represents the GECO score for a single iteration at the given k-value and the grey lines are iterations with random genes selected and scored as pseudo ribosomal protein genes.
(TIF)

**S5 Fig. Overlap between k91 DICE clusters and WGCNA clusters.** The x-axis shows the cluster numbers from the k91 DICE clusters, while the y-axis shows the cluster numbers from the WGCNA clusters. **A**. The overlap of genes from the sex-specific (Y-chromosome) ground truth set between the k91 DICE clusters and the WGCNA clusters. All sex-specific genes were found within a single k91 DICE cluster. **B**. The overlap of genes from the ribosomal ground truth set between the k91 DICE clusters and the WGCNA clusters. **C**. The overlap of genes from the immunoglobulin ground truth set between the k91 DICE clusters and the WGCNA clusters.
(TIF)

**S6 Fig. Determining soft-threshold for WGCNA.** The WGCNA R package requires a user-defined soft-threshold (referred to as the "power" in the function call) based on visually observing the above graph; specifically, the scale-freeness. The first value above the threshold suggested by the WGCNA authors (0.95) was chosen as indicated by the graph. Combined with the negative slope indicated in the mean connectivity graph, we can assume that we have a scale-free network as required by WGCNA.
(TIF)

**S1 Table. Ground truth gene sets.** The ground truth genes contained within each ground truth set.
(RTF)

**S2 Table. K-means derived GECO Scores.** The table of GECO scores generated after clustering with k-means using each value of k listed in the first column 'k'.
(CSV)

**S3 Table. WGCNA derived GECO Scores.** The table of GECO scores generated after clustering with WGCNA.
(CSV)

**S4 Table. The DICE Clusters.** The gene clusters derived from the DICE dataset created by the GECO metric.
(CSV)

**S5 Table. The ground truth genes' location within the k91 DICE clusters.** The concentration of ground truth genes within the k91 DICE clusters.
(XLSX)

**S6 Table. k91 DICE clusters scored with GECO using MSigDB gene sets.** The GECO scores for the k91 DICE clusters using the MSigDB hallmark gene sets as the ground truth genes.
(XLSX)

**S7 Table. The DICE Clusters' annotations.** For each DICE cluster 1–91, the most significantly enriched terms (adj. p-val < 0.05) for each DICE cluster are reported from GO ontologies molecular function (MF), biological process (BP), and cellular components (CC) as well as

KEGG pathways.
(XLSX)

**S8 Table. The whole-blood Tuberculosis signature.** The whole-blood TB signature from Berry et. al. [21].
(CSV)

## Acknowledgments

We thank Dr. Ky Sha (La Jolla Institute for Immunology) for comments on the manuscript and useful discussion. This content is solely the responsibility of the authors and does not necessarily represent the official views of the National Institutes of Health.

## Author Contributions

**Conceptualization:** Bjoern Peters.

**Formal analysis:** Jason Bennett, Mikhail Pomaznoy, Akul Singhania.

**Supervision:** Bjoern Peters.

**Writing – original draft:** Jason Bennett.

**Writing – review & editing:** Jason Bennett, Mikhail Pomaznoy, Akul Singhania, Bjoern Peters.

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
