## [Decision Letter · Decision Letter 0]

21 Jun 2021

Dear Dr. Peters,

Thank you very much for submitting your manuscript "A metric for evaluating biological information in gene sets and its application to identify co-expressed gene clusters in PBMC" for consideration at PLOS Computational Biology.

As with all papers reviewed by the journal, your manuscript was reviewed by members of the editorial board and by several independent reviewers. In light of the reviews (below this email), we would like to invite the resubmission of a significantly-revised version that takes into account the reviewers' comments.

Thank you for your patience. As you can see there are couple of concerns that the reviewers have identified. We would recommend addressing all of them, and if some are not possible, please provide a detailed justification in your response letter. In particular, revising the manuscript to address the clarity and the novelty issues better would be very important. We look forward to read your revised manuscript.

We cannot make any decision about publication until we have seen the revised manuscript and your response to the reviewers' comments. Your revised manuscript is also likely to be sent to reviewers for further evaluation.

Sincerely,

Tamas Korcsmaros

Guest Editor

PLOS Computational Biology

Sushmita Roy

Deputy Editor

PLOS Computational Biology

Thank you for your patience. As you can see there are couple of concerns that the reviewers have identified. I would recommend addressing all of them, and if some are not possible, please provide a detailed justification in your response letter. In particular, revising the manuscript to address the clarity and the novelty issues better would be very important. I look forward to read your revised manuscript.

Reviewer's Responses to Questions

**Comments to the Authors:**

Reviewer #1: Bennett et al described a really clever metric for clustering which they call GECO (Grund truth and evaluation of clustering outcomes). They used transcriptomic signatures such as ribosomal signatures, male gender-specific signature and an immunoglobulin signature to measure how well certain clustering methods cluster these genes together. I think the method itself is really clever. I have a few comments regarding the manuscript.

Mayor issues:

Probably the authors have forgotten to mention that they used the logarithmic count of the expression values. Using the expression values directly in a k-mean clustering will make the absolute values dive the clustering and not which genes are co-expressed, have similar gene expression pattern in the samples. I have read that they used the TPM normalised Z scores, but I am not sure whether the Z scores are normalised gene-wise or sample wise. They should be normalised gene-wise to be comparable. Please add this or use log(TPM) counts, because those are closer to a normal distribution, where k means is applied.

My other issue is the comparison with WGCNA. The authors optimised only one parameter of the WGCNA analysis (deepSplit parameter). WGCNA is sensitive to the power of the correlations are used. Please show the graphs of the scale-freeness of choosing power 10 as a supplementary figure. Also in WGCNA what was the actual input data - Z score normalised gene expression or TPM? I suggest adding a heatmap that compares the WGCNA, k-means and k-91 DICE clustering similarity using the various datasets - overlap percentage in the squares of the heatmap.

On the gene expression analysisi what background was used? Please describe it in the methods.

Minor:

Please check whether all acronyms are defined. I am not sure that TB is defined in the introduction or in the results.

Reviewer #2: The main problem in analyzing transcriptomics data is that the number of independent samples is typically much lower (<100)

than the number of genes whose expression is quantified, particularly for the single cell RNA-seq.

The authors to design a novel metric for the evaluation of clustering of genes based on the gene expression profiles.

In all, the manuscript is well-organized and easy to follow. However, I found some many problems:

1. The authors claim that the proposed GECO metric is computational efficient, robust and superior to the current alternatives.

As shown the Section GECO metric (Line: 137), the proposed method only exploit the overlapping genes between the predicted cluster

and ground truth one, which is very similar to the classic metric, such as Jaccard index, and hypergemotric test. Thus, I doubt

the novelty of metric. I notice that the subtle difference exists, how the gentle modification of the available alternatives may not

be very interesting.

2. One issue really confused me is that the authors adopt k-means to validate the performance of the proposed metric. Actually, k-means is

criticized for the instability, sensitivity to the initial solution, and low accuracy. There are some many algorithms can be used, such

as Hierichical clustering, spectral clustering, principle component analysi, and nonnegative matrix factorization, which significantly

outperforms k-means. Why do not you take these ones?

3. In the experiments, the authors only employ three gene sets to validate the performance, which is insufficient to validate the performance.

Furthermore, the biological story is not very interesting.

Based on the consideration above, I cannot recommend it to be published.

Reviewer #3: Reviewer Comments:

The authors of this manuscript highlight and address important drawbacks of current gene cluster/module identification from gene expression data - namely that unbiased reduction of gene expression into clusters/modules may identify ones that discriminate among samples based on confounding factors, and that there are no methods to determine the biologically optimal number of clusters/modules. The authors note the existing disconnect between the technical principles at play in determining “optimal” clustering parameters from a mathematical perspective, and the necessity to cluster genes into clusters that illustrate true biological signal in gene expression. They propose and describe a method that uses sets of consistently/canonically co-expressed sets of genes to inform and “ground-truth” the module building process; potentially improving module building from new datasets by aiding the user in selecting technical parameters such that resulting modules are optimized for their biological relevance. From this method, they develop a set of 91 genesets that can be used in particular for immune cell type specific expression profiling experiments. Overall, their approach is sound and the method represent a valuable contribution to the field. However, I have several suggestions to clarify and potentially strengthen the manuscript.

Major Comments:

The manuscript would be strengthened by acknowledging/exploring potential drawbacks of the proposed method (there are currently few caveats mentioned). Ie, could this method risk simply confirming already described expression profiles/clusters by using those existing gene sets to “ground truth” findings? Similarly, could it risk fine tuning mathematical parameters until finding a predefined signal for a given dataset (which presumably could occur if disease specific gene sets were used as the “ground truth”)? How does the method avoid these types of issues? While I do not believe the method necessarily would result in these drawbacks, the manuscript should head off these concerns by discussing them.

To this end, further defining guidelines for the selection/curation of appropriate ground-truth gene sets would be valuable (and standardized way to first test for appropriate co-expression of ground-truth gene sets within the data at hand). To what extent does responsibility fall purely on the user to define/test these ground-truth gene sets?

While the manuscript is generally well written and clearly worded, there are certain areas that lack clarity. For example it is often difficult to discern which underlying test dataset the authors are discussing, as it varies from section to section within the results and is sometimes not stated. Furthermore, the paper presents a substantial amount of methods information in the results section, which makes is difficult to readily gain comprehensive understanding of their process while reading through the manuscript. Some of the results section would be more appropriately placed in the introduction or methods, and some restructuring suggestions are made below.

The current manuscript focuses on 3 ground truth gene sets that represent canonical biology. The authors extend the work to show the potential value with a more context specific dataset using TB gene set signatures as ground truth comparisons. To the earlier point about appropriate selection of ground truth gene sets, the authors may want to consider extending this to other signatures that have been well characterized – eg. interferon stimulated genes, canonical signaling pathways, MSigDB Hallmark gene sets. This may strengthen the manuscript for a broad audience.

Evaluation and comparison of two clustering methods (k-means and WGCNA) is an important asset of this manuscript. However, WGCNA is not mentioned until the middle of the methods section (and there seems like a bit of an afterthought). The authors later include valuable discussion comparing the two methods and assessing the potential utility of the GECO approach into both. Including some of this discussion in the introduction as it increases the relevance of the manuscript and provides additional justification for the new method proposed.

As noted above, the manuscript as presented will benefit from some restructuring. Currently the methods and results section are somewhat poorly distinguished, with many relevant methods details not given until the results section (e.g. assessment of ground-truth gene sets for co-expression in the analysis dataset, multiple iteration of k-means clustering to account for stochasticity, gene set enrichment analysis). This leaves the reader desiring more detail that is then not given until later in the manuscript. The overview given under “Establishment of the GECO Metric” in the results section would be a helpful outline of the methods if included at the end of the introduction or beginning of the methods section. I would recommend beginning the methods section with this sort of brief but comprehensive overview of the analytic process (expanding conceptual figure 1 and citing it in this section), before giving specific methodological detail in each step. This journal generally requests a “results & discussion before methods” format, and the authors could consider restructuring their manuscript in this way.

Methods - GECO Metric This section needs work adding clarity to the description of how the metric is derived. It may be clearer to define the mathematical calculations first, and then describe their intuitive interpretation. This section could also be better organized to avoid repeating the calculations. The calculations presented in Figure 1 are a much more intuitive way of describing the approach than is given in the text. I would make the description of the calculation more like that given in the figure. Figure 1 could be cited sooner, in the methods section as it is helpful to understand the workflow. The ROC method used to generate the GECO score should be better explained, both in terms of what this method technically calculates, and an intuitive explanation of what this information tells the user (ie, why is this the optimal way to derive the metric). I believe that this is more or less and integrative method to evaluate the ability to predict co-occurrence of genes from a ground-truth set within modules, but this lacks clarity. It seems that there should also be a description of how gene scores are used to generate predictions on which the ROC curves are fitted. Figure 1 helps explain how the gene scores described in this section are used to produce the GECO metric via ROC (though it seems a prediction step should be added), reference it.

Minor comments/Line Notes:

L277-288 Description of evaluating the GECO metric is described first and in detail only for the ribosomal gene set. This is somewhat confusing to the reader and would be clearer to describe the general process & then state that it was conducted for each of three gene sets and present the results.

L301 – the listed k values for WGCNA “49 and 46” are in reverse order of what they should be for deepsplit 3 vs 4.

L328 Clarify that the “74-gene TB signature” is derived from the Burel dataset. Is this not problematic to ground-truth modules built from the Burel data using a module previously described from that same dataset? This seems circular.

L340 “We selected the most representative of those iterations”

What does this mean? A better description of how the final gene sets were chosen

should be included here or in the methods, this would be relevant to using the approach in other datasets.

L281 More description should be included (in the methods) about how random gene sets were generated. Ie how many randomized gene sets, how large, are they non-overlapping? Etc.

L359-375 It is not clear (or stated) in which dataset this evaluation is being conducted.

L359-375 It is not clear in which RNASeq dataset these comparisons are being made. I would assume these comparisons are made using the Burel dataset (since a CD4 TB signature would not be expected in the healthy donor DICE data), but this is not stated.

Figures/Tables

Supplementary tables should more appropriately refer to biological “sex” as opposed to “gender”, consistent with terminology used throughout the manuscript.

**Have the authors made all data and (if applicable) computational code underlying the findings in their manuscript fully available?**

Reviewer #1: Yes

Reviewer #2: Yes

Reviewer #3: None

PLOS authors have the option to publish the peer review history of their article (what does this mean?). If published, this will include your full peer review and any attached files.

Reviewer #1: **Yes: **Dezso Modos

Reviewer #2: No

Reviewer #3: No
---

## [Editor Report · Decision Letter 1]

17 Sep 2021

Dear Dr. Peters,

Thank you your revision, which fully addressed all the concerns that previously were raised by the three reviewers.

Therefore, we are pleased to inform you that your manuscript 'A metric for evaluating biological information in gene sets and its application to identify co-expressed gene clusters in PBMC' has been provisionally accepted for publication in PLOS Computational Biology.

Best regards,

Tamas Korcsmaros

Guest Editor

PLOS Computational Biology

Sushmita Roy

Deputy Editor

PLOS Computational Biology

---

## [Editor Report · Acceptance letter]

4 Oct 2021

PCOMPBIOL-D-21-00672R1 

A metric for evaluating biological information in gene sets and its application to identify co-expressed gene clusters in PBMC

Dear Dr Peters,

I am pleased to inform you that your manuscript has been formally accepted for publication in PLOS Computational Biology. Your manuscript is now with our production department and you will be notified of the publication date in due course.

With kind regards,

Agnes Pap
